# Selective Liquid-Phase Oxidation of Toluene with Molecular Oxygen Catalyzed by Mn$_3$O$_4$ Nanoparticles Immobilized on CNTs under Solvent-Free Conditions

**Yuwei Feng and Aiwu Zeng \***

State Key Laboratory of Chemical Engineering, School of Chemical Engineering and Technology,
Tianjin University, Tianjin 300350, China; fengyw@tju.edu.cn
\* Correspondence: awzeng@tju.edu.cn; Tel.: +86-155-2207-2156

**Abstract:** The catalytic performance of Mn$_3$O$_4$ supported on carbon nanotubes (CNTs) in the liquid-phase oxidation of toluene to benzyl alcohol and benzaldehyde was studied. The supported catalysts were characterized by X-ray diffraction (XRD), Raman spectroscopy, scanning electron microscopy (SEM), transmission electron microscopy (TEM), X-ray photoelectron spectroscopy (XPS), N$_2$ adsorption–desorption isotherms and ICP-MS. The results demonstrate that Mn$_3$O$_4$ nanoparticles loaded on CNTs performed better compared with pristine Mn$_3$O$_4$ or CNTs. The main reason for the increased catalytic activity is the dispersion and loading of Mn$_3$O$_4$ in CNTs. By optimizing the reaction temperature, reaction time, catalyst quality, oxygen flow rate and initiator dosage, the optimum reaction conditions were obtained. Using tert-butyl hydroperoxide (TBHP) as the initiator and oxygen as the oxidant, the toluene conversion rate was as high as 24.63%, and benzyl alcohol and benzaldehyde selectivity was 90.49%. The good stability of the catalyst was confirmed by repeating the experiment for four cycles and observing no significant changes in its performance.

**Keywords:** toluene; benzyl alcohol; benzaldehyde; selective oxidation; Mn$_3$O$_4$; carbon nanotubes

## 1. Introduction

The selective oxidation of toluene is an essential type of catalytic oxidation reaction, which results in benzyl alcohol, benzaldehyde, and benzoic acid as reaction products. These chemical products are extensively used in different industries and processes. For example, benzyl alcohol is widely utilized in the chemical, food, medicine, ink, printing, and dyeing and photosensitive industries [1], and has extensive prospects. The traditional synthesis process of benzyl chloride is through hydrolysis, which has been widely criticized for its severe pollution, equipment corrosion, and chlorine-containing products. With the development of green and sustainable chemistry, the toluene side-chain oxidation method has attracted the attention of many scholars [2]. However, such reaction requires large energy consumption for the bond breakage of the toluene side chain [3]. As a consequence, the reaction is usually carried out under high temperature and pressure conditions, which results in undesired esters and Co$_x$ products due to over-oxidation, and reaction control is challenging.

Studies on the selective oxidation of toluene to benzaldehyde have recently made great progress regarding catalyst synthesis and mechanism research [4–6]. However, improvements can be made to increase the selectivity of benzyl alcohol. Although the selectivity of benzyl alcohol and benzaldehyde reached 60%, the conversion rate was only 8.9% [7]. Sun Wen [8] prepared an F-modified CuNiAl hydrotalcite catalyst and found that the selectivity of benzyl alcohol in the toluene-H$_2$O$_2$ oxidation system was as high as 81.4%, but toluene conversion rate was only 8.4%. Surprisingly, toluene

conversion rate was only 0.1%, with oxygen under the same reaction conditions. Surajit [9,10] et al. designed and synthesized a Schiff base copper catalyst containing an NO ligand. The toluene-$H_2O_2$ oxidation system showed a good catalytic performance, in which toluene conversion rate was 82.0% and benzyl alcohol yield was 66.0%. The bond length of the Schiff base ligand of the catalyst was calculated by DFT, and it is believed that the NO ligand of the catalyst cannot only stabilize the benzyl hydroperoxide intermediate through hydrogen bonding, but also reduce the free radical reaction rate to improve benzyl alcohol selectivity. John [11] et al. synthesized $Mn_3O_4$ supported on the hydrophobic carrier DVTA by a solvothermal method and applied it to the selective oxidation of toluene. The newly studied hydrophobic catalyst exhibited both benzyl alcohol and benzaldehyde in the toluene–oxygen reaction system. At the same time, tert-butyl hydroperoxide as the initiator played a vital role in the reaction. Finally, toluene conversion rate was 34.0%, and the selectivity to benzyl alcohol was 45.7%. Selective oxidation reaction is greatly limited by certain oxidants, such as $H_2O_2$ and TBHP. This not only increases production cost but also impacts catalyst separation and use. Therefore, it is indispensable to prepare a highly active, simple, and recyclable catalyst for the oxidation of toluene in oxygen.

Compared with their homogeneous counterparts, heterogeneous catalysts have the advantages of easy product separation, catalyst reusability, and easy oxygen activation to accelerate the reaction process [12]. The transition metal manganese oxide has high activity and reducibility due to its variable valence electron pair. This oxide has been widely used in selective oxidation studies. Some examples include $MnCO_3$-catalyzed oxidation of toluene to benzoic acid [13], $Mn_3O_4$-catalyzed selective oxidation of toluene [14–16], and $MnO_2$-catalytic oxidation of toluene [17]. Carbon nanotubes (CNTs), as a class of carbon nanomaterials, have excellent physical and chemical properties such as strong corrosion resistance, good thermal stability, electronic modification, high specific surface area and multiple layers [18]. CNTs have shown good application prospects as catalysts [19,20] or supports [21,22] and have been widely used in the catalysis of carbon monoxide [23], hydrocarbons [24,25], and alcohol [26]. Studies have shown that structural defects in carbon nanotubes can improve the binding ability of metals or metal oxides on the catalyst surface and increase catalytic activity [27,28]. For example, Song [27] reported the CuO/CNT selective catalytic oxidation of ammonia gas to produce $N_2$ and obtained a $N_2$ selectivity of 98.7%. Carbon nanotubes have been reported in the liquid-phase selective oxidation of ethylbenzene to acetophenone [29]. It has been proven that CNTs as non-metallic catalysts have an important role in the selective formation of decomposition reaction intermediates, PEHP, and reaction products. Doping and modifying metal catalysts with CNTs can not only prevent $Mn_3O_4$ from agglomerating, but can also activate metal oxides [30], promote electron transfer [31,32], and improve catalytic capacity.

In this study, the one-step solvothermal synthesis of $Mn_3O_4$/CNTs catalysts was applied to the selective oxidation of toluene. The catalytic effect that the concentration of CNTs doped on the manganese oxide-supported catalyst has on the liquid-phase catalytic oxidation of toluene was studied. Finally, experimental results demonstrated that the supported catalyst showed a high activity and selectivity in the oxidation of toluene to benzyl alcohol and benzaldehyde.

## 2. Results and Discussion

### 2.1. Catalyst Characterization Analysis

Both $Mn_3O_4$, and $Mn_3O_4$/CNTs-3 catalysts were characterized by XRD, as shown in Figure 1. It can be observed from the figure that the catalyst has diffraction peaks at $2\theta = 18°$, $28°$, $32°$, $36°$, among others. By comparing the obtained spectra to the standard JCPSD file card no. 24-0734 [33], it was found that the diffraction peaks at $18.1°$, $28.9°$, $31.1°$, $32.3°$, $36.1°$, $38.0°$, $44.5°$, $50.7°$, and $59.8°$ correspond to (101), (112), (200), (103), (004), (220), (211), (105), and (224) crystal planes, respectively. Additionally, the (211) crystal plane corresponding to the characteristic peak at $36.1°$ proves that the active component of the catalyst is $Mn_3O_4$. No diffraction peaks of other manganese oxide crystal

phases were found, proving that $Mn_3O_4$ is of high purity. A new diffraction peak at 26.1° was observed for the supported $Mn_3O_4$/CNTs catalyst, contrary to the unsupported $Mn_3O_4$. It was found that the peak belongs to the graphite phase, corresponding to the (002) crystal plane (JCPSD file card no. 41-1487), which confirms that the supported catalyst is doped with CNTs. The grain size of $Mn_3O_4$ deposited on CNTs, calculated according to the Scherrer formula, was between 3 and 10 nm and that of pure $Mn_3O_4$ was 15 nm. This may be because the addition of carbon nanotubes promotes heterogeneous nucleation [34].

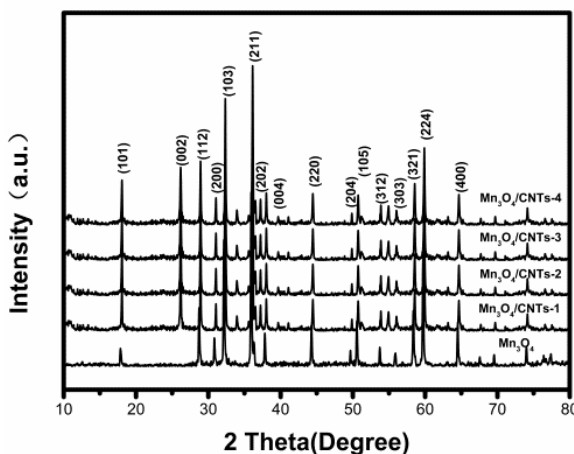

**Figure 1.** XRD patterns of the $Mn_3O_4$ and $Mn_3O_4$/CNTs catalysts composites.

Figure 2a,b show the SEM images of CNTs and $Mn_3O_4$/CNTs-3. Comparing the two images reveals a different diameter for the carbon nanotubes for each of the catalysts. Additionally, from the $Mn_3O_4$/CNTs-3 TEM in Figure 2c, it can be determined that the difference in diameter is due to the fact that the $Mn_3O_4$ nanoparticles are uniformly supported on the surface of the CNTs. The diameter of the $Mn_3O_4$ nanoparticles is approximately 3 to 5 nm. The HRTEM of the loaded component reveals that the clear fringe spacing is 0.48 nm, which corresponds to the (101) crystal plane of octahedral $Mn_3O_4$. This suggests that $Mn^{7+}$ is reduced and converted into $Mn^{3+}$ and $Mn^{2+}$ by ethylene glycol under hydrothermal reaction conditions, which results in tetrahedral and octahedral holes occupied by $Mn^{2+}$ and $Mn^{3+}$ ions, respectively [35]. $Mn^{3+}$ ions are more easily generated in the (101) direction, which further promotes the growth of the lattice plane in the (101) direction to form octahedral $Mn_3O_4$. The exposed (101) crystal plane contains more active Mn–O bonds, which is conducive to the adsorption and activation of the reactants and promotion of the reaction [36].

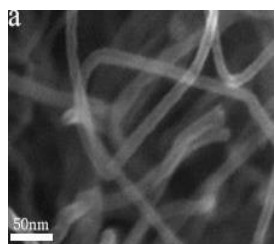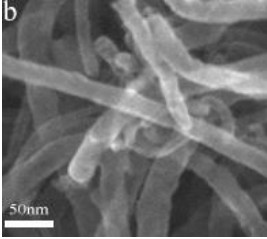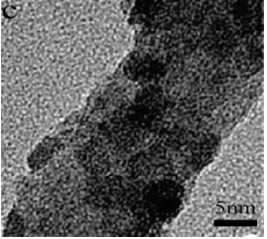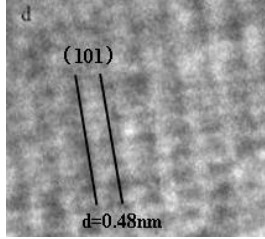

**Figure 2.** SEM image of (**a**) purified CNTs (**b**) $Mn_3O_4$/CNTs-3 composites; (**c**) TEM image of $Mn_3O_4$/CNTs-3; (**d**) HRTEM image of the $Mn_3O_4$ supported on CNTs.

Raman spectroscopy was used to determine the degree of structural order or the presence of defects in CNT materials. A Raman spectroscopy analysis was carried out on the four supported catalysts, as shown in Figure 3. In addition to the $Mn_3O_4$ peak observed at 659 $cm^{-1}$ [37], a D-band reflecting the disordered graphite structure was observed at 1338 $cm^{-1}$ while a G-band reflecting the structure of $sp_2$ hybridized carbon atoms was observed at 1576 $cm^{-1}$ [38]. In order to measure the

degree of defects of the carbon nanotubes in the catalyst, the following expression was used: R = $I_D/I_G$. Carbon nanotube defect sites often affect active component loading. When the loading of CNTs in the catalyst increased from 40 to 80 mg, the R value decreased from 1.13 to 0.95. When the loading of CNTs in the catalyst was 100 mg, the R value was 1.06. These results prove that the defects on the CNTs are affected by the metal oxide loading, which is consistent with previous reports [28].

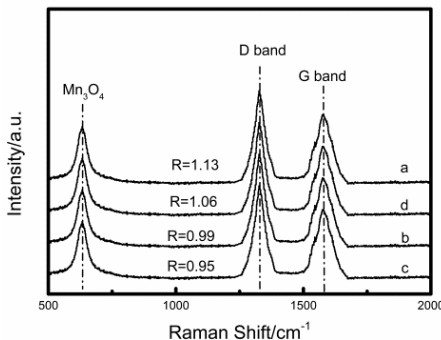

**Figure 3.** Raman spectra of (**a**) $Mn_3O_4$/CNTs-1 CNTs (**b**) $Mn_3O_4$/CNTs-2 (**c**) $Mn_3O_4$/CNTs-3 (**d**) $Mn_3O_4$/CNTs-4.

The BET surface area, pore volume, and pore diameter of the supported catalyst are shown in Table 1. Compared with the specific surface area of the carrier CNTs of 341 $m^2$/g, the specific surface area of the four supported catalysts $Mn_3O_4$/CNTs was reduced. This indicates that the hydrothermally synthesized $Mn_3O_4$ deposited on the surface of the support, resulting in a decrease in specific surface area. This conclusion can also be obtained from the TEM and Raman analyses. However, as the amount of carbon nanotubes increases, the pore volume and pore diameter increase. Since the surface of the CNTs has more defect sites, an interaction occurs between the carbon nanotube framework and the metal oxide by loading to prevent metal oxide agglomeration [39,40]. This is beneficial to the distribution of active components in defects of the CNTs and improves the redox ability. Through comparison of activity data, it was found that large specific surface area and pore volume are effective for catalytic activity. Mn content of the four supported catalysts was analyzed by ICP-MS to be 7.1%, 8.2%, 8.9%, and 6.4% (wt %) for samples loaded with 40, 60, 80, and 100 mg of CNTs, respectively. The addition of excess carbon nanotubes may restrict the homogeneous nucleation growth of $Mn_3O_4$ under hydrothermal conditions and prevent the catalyst from being fully loaded on the carrier. This may be the reason for the reduced catalytic activity.

**Table 1.** BET surface area of CNTs and $Mn_3O_4$/CNTs catalysts.

| Catalyst Composites [a] | Notation | CNTs [b] | BET Surface Area ($m^2$/g) | Pore Volume ($cm^3$/g) [c] | Pore Diameter (nm) [d] |
|---|---|---|---|---|---|
| CNTs | CNTs | - | 341 | 1.2321 | 14.5 |
| $Mn_3O_4$/CNTs(1:0.8) | $Mn_3O_4$/CNTs-1 | 0.4 | 251 | 0.7842 | 12.0 |
| $Mn_3O_4$/CNTs(1:1.2) | $Mn_3O_4$/CNTs-2 | 0.6 | 268 | 0.8136 | 12.1 |
| $Mn_3O_4$/CNTs(1:1.6) | $Mn_3O_4$/CNTs-3 | 0.8 | 269 | 0.8421 | 12.5 |
| $Mn_3O_4$/CNTs(1:2.1) | $Mn_3O_4$/CNTs-4 | 1 | 272 | 0.9031 | 13.2 |
| $Mn_3O_4$/CNTs-run | $Mn_3O_4$/CNTs-3-run4 | - | 270 | 0.8532 | 12.6 |

[a] The mass ratio of $Mn_3O_4$/CNTs estimated from reactants; [b] The mass percentage of added CNTs in preparation process; [c] BJH (Barret-Joyner-Halenda)adsorption average pore volume; [d] BJH adsorption average pore diameter.

The XPS analysis of $Mn_{2p}$ and $O_{1s}$ of $Mn_3O_4$/CNTs nanoparticles is summarized in Figure 4A,B. The peak at 641.2–642.1 eV corresponds to $Mn2p_{3/2}$, while the peak observed at 653–653.9 eV corresponds to $Mn2p_{1/2}$. The spin splitting energy can be determined as 11.8 eV, corresponding to the Mn $2p_{1/2}$ and Mn $2p_{3/2}$ spin-orbit states of $Mn_3O_4$. In addition, due to the interaction between Mn and the CNTs, the Mn 2p shifts to a lower binding energy for $Mn_3O_4$/CNTs-3 compared with $Mn_3O_4$/CNTs-1 and

Mn$_3$O$_4$/CNTs-2, indicating that Mn$_3$O$_4$ species have been doped in CNTs. In the XPS spectrum of O 1s, the binding energy of O 1s shifts towards the direction of lower binding energy as the load increases. This suggests that the doping of carbon nanotubes increases the electron cloud density of the Mn=O bond on the catalyst surface.

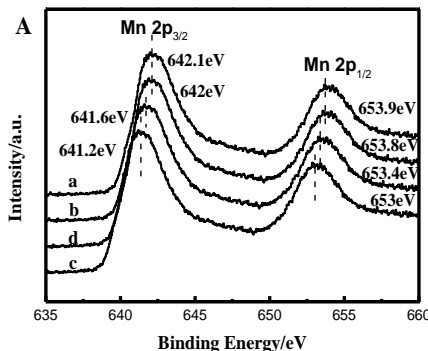 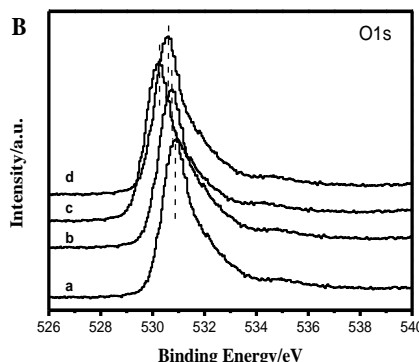

**Figure 4.** XPS spectra of (**A**) Mn$_{2p}$ and (**B**) O$_{1s}$. (a) Mn$_3$O$_4$/CNTs-1 CNTs, (b) Mn$_3$O$_4$/CNTs-2, (c) Mn$_3$O$_4$/CNTs-3, (d) Mn$_3$O$_4$/CNTs-4.

### 2.2. Results of Catalyst Performance Evaluation of Toluene Liquid-Phase Catalytic Oxidation Reaction

In this study, the main products of the reaction are benzyl alcohol and benzaldehyde, as well as a small amount of benzoic acid and esters. The results of the final catalytic oxidation experiment are shown in Table 2. When the catalyst was only Mn$_3$O$_4$ or CNTs, toluene conversion rate is low, and the catalytic oxidation reaction does not occur. This indicates that the defect sites of the CNTs are not catalytic oxidation active centers. When adding CNTs to Mn$_3$O$_4$, it was found that the catalytic activity of Mn$_3$O$_4$ was significantly improved, and the selective oxidation of toluene had a significant promotion effect. When the content of CNTs in the Mn$_3$O$_4$ catalyst was increased from 40 to 80 mg, toluene conversion rate increased from 7% to 24.63%. It can be seen from the characterization that the increase in specific surface area and the dispersion of Mn$_3$O$_4$ may explain the increased catalytic activity. Moreover, it has been determined that the benzene ring in toluene can be selectively adsorbed with CNTs through π electrons, which provides a possible way to accelerate the activation of toluene [41,42]. The data presented here also reveal that benzyl alcohol selectivity decreased with increasing catalytic activity. This is because the benzyl alcohol produced by the reaction continues to oxidize to form benzaldehyde and benzoic acid, resulting in a decrease in selectivity. When the content of CNTs in the supported catalyst increased to 100 mg, the catalytic activity decreased. This may be due to the fact that the surface of the carbon nanotubes contains acidic groups such as carboxylic groups, which hinder the activated adsorption of toluene at the active site, leading to a reduced catalytic activity [43]. Luo [29] et al. reported that surface carboxylic groups on CNTs were unfavorable to saturated hydrocarbon oxidation.

**Table 2.** Catalytic activity of Mn$_3$O$_4$/CNTs catalyst on toluene oxidation reaction.

| Catalysis | Conversion/% | Selectivity/% | | |
|:---:|:---:|:---:|:---:|:---:|
| | | Benzyl Alcohol | Benzaldehyde | Other Products |
| Mn$_3$O$_4$ | 3.54 | 40.47 | 58.51 | 1.02 |
| Mn$_3$O$_4$ [a] | 1.67 | 38.57 | 61.19 | 0.24 |
| CNTs | 0.65 | 41.56 | 58.44 | 0 |
| Mn$_3$O$_4$/CNTs-1 | 7.00 | 58.14 | 40.59 | 1.27 |
| Mn$_3$O$_4$/CNTs-2 | 15.10 | 53.06 | 43.58 | 3.36 |
| Mn$_3$O$_4$CNTs-3 | 24.63 | 43.51 | 46.98 | 9.51 |
| Mn$_3$O$_4$/CNTs-4 | 18.56 | 50.52 | 44.03 | 5.45 |

[a] Commercial catalyst. Reaction conditions: 10 mL toluene, 100 mg catalyst, 0.5 mL TBHP (t-butylhydroperoxide), reaction temperature 90 °C, oxygen flow rate 15 mL/min, reaction time 12 h.

Reaction temperature is one of the key factors influencing the selective oxidation of toluene [44]. Therefore, the relationship between toluene conversion and product selectivity of $Mn_3O_4$/CNTs-3 catalyst was investigated at a temperature range between 70 and 100 °C. Toluene conversion rate increased as the reaction temperature increased, as observed in Figure 5. However, benzyl alcohol decreased continuously as reaction temperature increased. When the experimental temperature rose to 100 °C, the ineffective decomposition rate of tert-butyl hydroperoxide was accelerated and the radical reaction rate decreased, resulting in a slower growth rate of toluene conversion. At this time, the rate of oxidation of benzyl alcohol and benzaldehyde to benzoic acid was accelerated. Therefore, in the final experiment, benzyl alcohol yield was the highest when the reaction temperature was 90 °C.

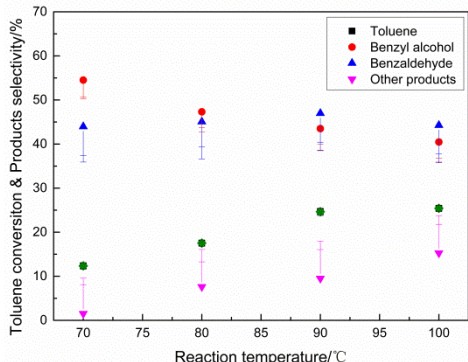

**Figure 5.** Catalytic performance for the selective oxidation of toluene on the $Mn_3O_4$/CNTs-3 at different temperatures.

As shown in Figure 6, the change in toluene conversion and product selectivity was analyzed in terms of reaction time for the different reaction products. In the initial stage of the reaction, toluene conversion was maintained at a relatively low level, and the reaction product was mainly benzyl alcohol. With increasing reaction time, the number of benzyl radicals generated by the activation of toluene by tert-butyl hydroperoxide increased, resulting in an increase in reaction rate. At this time, benzyl alcohol oxidized to form benzaldehyde and benzoic acid. When the reaction proceeded for 8 h, the benzyl alcohol production rate was lower than that of benzaldehyde. When the reaction progressed for 12 h, benzaldehyde selectivity began to decrease, and toluene began to be deeply oxidized to generate benzoic acid.

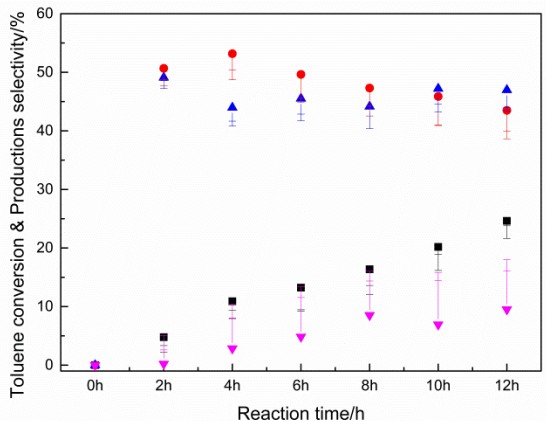

**Figure 6.** Catalytic performance for the selective oxidation of toluene on the $Mn_3O_4$/CNTs-3 with different reaction times.

Additionally, the effect of oxygen flow rate on the catalytic oxidation of toluene was investigated and the results are shown in Figure 7. When the oxygen flow rate was lower than 5 mL min$^{-1}$, toluene conversion rate was low, approximately 6.8%. This low conversion rate may be due to the fact that the

mass transfer resistance in the gas–liquid–solid reaction system limits the oxidation reaction of toluene on the catalytic oxidation surface [45]. Increasing the oxygen flow rate overcomes such mass transfer resistance. When the oxygen flow rate continued to increase to 15 mL min$^{-1}$, toluene conversion rate was significantly improved, likely due to the improvement of the amount of active oxygen participating in the oxidation of the catalyst surface, which shows that oxygen plays an important role in this reaction. However, if oxygen flow rate was increased to 20 mL·min$^{-1}$, benzyl alcohol selectivity was extremely reduced. This may be because the benzyl alcohol formed on the surface of the catalyst can continue to be oxidized in the oxygen atmosphere to form benzaldehyde and benzene formic acid.

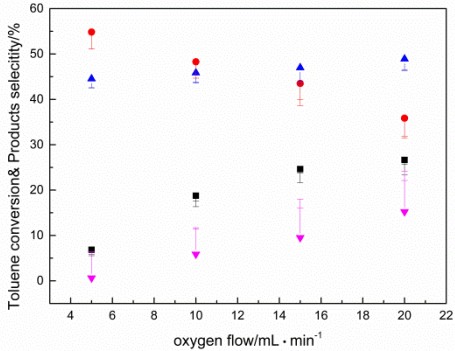

**Figure 7.** Catalytic performance for the selective oxidation of toluene on the Mn$_3$O$_4$/CNTs-3 with different oxygen flow rates.

The number of active sites on the catalyst surface significantly influences the selective oxidation of toluene in the catalytic reaction. The influence of the amount of catalyst on the catalytic oxidation reaction is shown in Figure 8. As the amount of catalyst increased, the catalytic reaction rate increased significantly. The catalytic reaction rate is affected by the catalytic active site. When the amount of catalyst added is 80 mg, due to the limitation of the catalytic active site, only part of the toluene can undergo catalytic reaction at the active site, which results in a low conversion rate. However, when the catalyst exceeds 100 mg, toluene conversion rate did not show any significant differences, but new oxidation products such as benzoic acid appeared; comprehensive consideration is needed to determine the amount of catalyst to 100 mg.

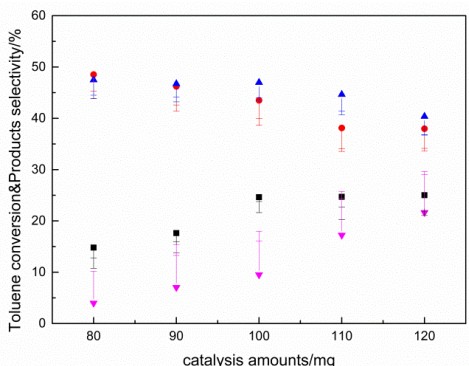

**Figure 8.** Catalytic performance for the selective oxidation of toluene on the Mn$_3$O$_4$/CNTs-3 with different amounts of catalysis.

In order to study the influence of tert-butyl hydrogen peroxide as an initiator in the selective oxidation reaction of toluene, the amount of participating tert-butyl hydrogen peroxide was varied when other reaction parameters were unchanged, and the results are shown in Figure 9. It was found that toluene oxidation did not occur in oxygen atmosphere when tert-butyl hydrogen peroxide was not added to the reaction process. However, when 0.1 mL of tert-butyl hydrogen peroxide were

added to the reaction, toluene conversion rate increased from 0% to 7.4%. Nevertheless, when the tertiary butyl peroxide hydrogen content increased to 1 mL, toluene conversion rate was observed to flatten or even decrease. This phenomenon could be due to the fact that tertiary butyl peroxide decomposition generated tert-butyl alcohol after adsorption to the surface of the catalyst and competed with toluene adsorption, which in turn limited toluene to continue to spread in the catalyst surface, eventually resulting in the decrease in toluene conversion rate [2]. At the same time, benzyl alcohol and benzaldehyde produced by the reaction began to further oxidize to benzoic acid and other substances, resulting in a decrease in alcohol and aldehyde selectivity.

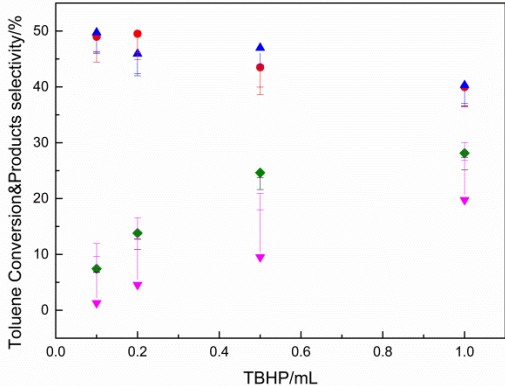

**Figure 9.** Catalytic performance for the selective oxidation of toluene on the $Mn_3O_4$/CNTs-3 with different TBHP.

Lifetime is one of the important factors to evaluate the performance of a catalyst. Catalyst recycling was investigated under optimal reaction conditions. After the reaction occurred, the catalyst was centrifugally filtered and simply washed and dried to be recovered. Then, toluene was catalyzed under the same reaction conditions to detect and analyze the reaction products. According to the experimental results, as shown in Table 3, the catalytic activity was not significantly changed after four cycles of use.

**Table 3.** Recyclability test for catalyst $Mn_3O_4$/CNTs-3 in the selective oxidation of toluene.

| Run | Conversion/% | Selectivity/% | | |
| --- | --- | --- | --- | --- |
| | | **Benzyl Alcohol** | **Benzaldehyde** | **Other Products** |
| 1 | 24.63 | 43.51 | 46.98 | 9.51 |
| 2 | 23.76 | 44.53 | 46.72 | 8.75 |
| 3 | 23.53 | 42.56 | 48.41 | 9.03 |
| 4 | 22.97 | 42.65 | 48.39 | 8.96 |

## 3. Materials and Methods

### 3.1. Materials

Toluene, potassium permanganate, methyl alcohol, and ethanol were obtained from Tianjin Yuanli Company (Tianjin, China). Benzyl alcohol, benzaldehyde, and benzoic acid were purchased from Shanghai Aladdin Chemical Reagent Company (Shanghai, China). Ethylene glycol was supplied by Real-Times Biotechnology Co., Ltd. (Beijing, China). Purified multi-wall carbon nanotubes (CNTs) with a diameter of 10–20 nm were sourced from Chengdu Organic Chemicals Co., Ltd. (Chengdu, China). Tert-butyl hydrogen peroxide was purchased from Kmart Tianjin Chemical Technology Co., Ltd. (Tianjin, China).

### 3.2. Catalyst Preparation

The supported catalyst $Mn_3O_4$/CNTs was prepared through a solvothermal method. The purified CNTs were dispersed in 30 mL of water at room temperature and sonicated for 30 min. Then, 100 mg of $KMnO_4$ solid powder were added and the solution was stirred at room temperature for approximately 2 h. An amount of 5 mL of ethylene glycol was incorporated to the above solution as a surfactant [46] and stirred for 2 h. The obtained brown liquid was transferred to a high-pressure reaction kettle and heated to 160 °C for 6 h, until a solid was obtained. The solid was filtered and washed with methanol and deionized water twice, dried overnight in a drying cabinet, and the $Mn_3O_4$/CNTs catalyst was obtained (see Table 1).

### 3.3. Catalyst Characterization

An X-ray diffractometer (XRD, Bruker AXS Co. Ltd., Karlsruhe, German, D8-focus, tube voltage 40 kV, current 40 mA, scanning range 10°–80°, sweep speed $5° \cdot s^{-1}$, $CuK\alpha$, $\lambda = 0.1540598$ nm), a scanning electron microscope (SEM, S-4800, Tokyo, Japan, acceleration voltage 1 kv), and a high-resolution transmission electron microscope (HRTEM, JEOLJEM-2100F, Tokyo, Japan, acceleration voltage 200 kV) were used to observe the sample morphology and structure. An inductively coupled plasma emission spectrometer (ICP-MS, Agilent 7900, California, CA, USA) was used to determine the Mn content in the resulting samples. XPS was utilized to analyze the relative content ratio of each element on the sample surface. The specific surface area and pore structure parameters of the catalyst were measured with a physical adsorption instrument. High-purity $N_2$ was used as the adsorbate and backfill gas. The relative pressure $(P/P_0)$ when measuring the pore volume was 0.99. The specific surface area was determined by the Brunauer–Emmett–Teller (BET) formula. LabRAM HR Evolution Raman spectroscopy (HORIBA) with a laser emission line of 532 nm was used.

### 3.4. Evaluation of Catalyst Performance

In a 25 mL round bottom flask with a reflux condenser and magnetic stirring, 10 mL of toluene and 100 mg of catalyst were added. An appropriate amount of oxygen was passed into the reaction system, and it was stirred at the set temperature for 12 h. After the reaction was completed, it was cooled to room temperature and filtered to obtain the product and recover the catalyst. The product composition was determined by AgilentGC-7890A (California, CA, USA). The test conditions were as follows: DB-FFAP capillary column 30 m × 0.25 μm × 0.25 mm, helium (purity 99.999%) as the carrier gas, inlet temperature 200 °C, and a chromatographic column temperature program. The initial temperature was 50 °C, maintained for 3 min. The temperature was raised to 230 °C at a rate of 20 °C min$^{-1}$, maintained for 6 min, and the column flow rate was 1.46 mL·min$^{-1}$.

## 4. Conclusions

In this article, the $Mn_3O_4$/CNTs-supported catalyst was successfully synthesized by a solvothermal method. The performance of the selective oxidation of toluene by changing the content of CNTs was compared and discussed. The reaction temperature, reaction time, oxygen flow rate, catalyst quality, and initiator dosage were optimized. The following conditions were found to give the highest catalytic conversion rate of 24.63% and a benzyl alcohol and benzaldehyde selectivity of 90.49%: $Mn_3O_4$/CNTs mass ratio 1:1.6, reaction temperature 90 °C, reaction time 12 h, oxygen flow rate 15 mL/min, catalyst amount 100 g, and initiator 0.5 mL. These results are related to the big surface area provided by CNTs and the good $Mn_3O_4$ dispersion. These data help to further enhance the research regarding supported catalysts in the field of selective catalytic oxidation.

**Author Contributions:** Conceptualization, methodology, formal analysis, validation, and review, supervision, project management, A.Z. Experimental work and draft writing, Y.F. All authors have read and agreed to the published version of the manuscript.

**Funding:** This research received no external funding.

**Conflicts of Interest:** The authors declare no conflict of interest.

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
