# Peer review of "Selective Liquid-Phase Oxidation of Toluene with Molecular Oxygen Catalyzed by Mn3O4 Nanoparticles Immobilized on CNTs under Solvent-Free Conditions"

_catalysts, doi:10.3390/catal10060623_

Round 1

Reviewer 1 Report

The present manuscript demonstrated the catalytic performance of Mn3O4 supported on carbon nanotubes (CNTs) for liquid-phase oxidation of toluene to benzyl alcohol and benzaldehyde. Toluene oxidation reaction has been well studied in the past using homogeneous and heterogeneous catalysts to produce benzyl alcohol as it is more valuable than other products. Authors synthesised various compositions of Mn3O4/CNTs catalysts and analyzed them thoroughly by using suitable characterization techniques and found that structural defects in CNTs played major role in enhancing catalytic activity and combined selectivity of benzyl alcohol and benzaldehyde. However, their catalytic part and correlations are incomplete that needs to be significantly improved. Further, I have couple of concerns about data representation and explanations part. Authors should go through my below points and address them.

  • Usually, in case of supported metal catalyst, active metal (supported metal/metal oxide) loadings will be optimized on supports, not support loadings to active metal or metal oxide. But in the present study authors reported in opposite away where they studied/represented optimizing support loading (CNTs) instead to active metal (Mn3O4) which is confusing. Please check the line 166 and line 210.
  • Also in table 1; Check the subtitle of “Catalysis type to Catalyst type”. Further I recommend changing the representation Mn3O4/CNTs(-xxx%) to Mn3O4 (xxx%)/CNTs.   
  • Table 2 shows that catalytic activity increased in the order of Mn3O4@CNTs-80 > Mn3O4@CNTs-100 > Mn3O4@CNTs-60 > Mn3O4@CNTs-40. But the selectivity towards to benzyl alcohols following opposite trend where Mn3O4@CNTs-80 is lower than other catalyst systems. Please provide some explanation why the selectivity following opposite trend compare to activity data.
  • The drawback of the present work is no controlled experiments/reactions to improve the benzyl alcohols selectivity instead combined benzaldehyde and benzyl alcohol selectivity. Benzyl alcohol is the most important and useful chemical compare to other toluene oxidation products. Further producing benzyl alcohol selectively by controlling other oxidation products from toluene is challenging.
  • I recommend changing the labeling in figure 5 from “toluene to conversion"

Reviewer 2 Report

This is an interesting manuscript dealing with the selective liquid-phase oxidation of toluene with molecular oxygen using Mn3O4 nanoparticles immobilized on CNTs catalyst. The authors also reported the preparation method of catalyst and determination of structure of it. I think that Authors have done a great effort studying the performance of these type of catalysts but some additional information and/or characterization of the catalysts would improve the significance of this work. The authors should consider the following points:

  • Page 4, line 167, Raman measurements – the Authors write – “When the loading of CNTs in the catalyst is 100 mg, the R value is 0.99”, however, Figure 3 shows that for Mn3O4/CNTs-100 R value is 1.06. Authors must solve this problem.
  • How does the porosity change – the Authors should compare pore size distribution of catalysts with support if they write about the distribution of active components in pores – page 5, line 178. Authors should present the pore size distribution for support and catalysts.
  • Why the BET surface area of Mn3O4/CNTs-40 and Mn3O4/CNTs-60 catalysts were larger than Mn3O4/CNTs-80 catalyst – Table 1. For the last one the BET surface area was the lowest. Authors should explain this phenomena.
  • Figure 4 A, XPS analysis - Is the order (a, b, d, c) correct?
  • Have the Authors compare the performance of catalysts in this work with the catalytic activity of other catalysts reported in literatures?
  • The authors claimed that the catalytic performance of Mn3O4/CNTs supported catalysts depends on content of CNTs and textural properties of the supports. However, the well-structured and logical arguments on catalysis are missing. So the catalytic activity part should be clarified with clear explanation by mentioning whether textural property and/or pore structure play a role for the studied reaction.

There are few errors those need to be corrected as well. For example: Table 1 “Mn3O4” should be “Mn3O4”; Figure 3 “Ranman” should be “Raman”, p.9 line 302 – Referee 1 – “yin, S. z.”?

Reviewer 3 Report

Reviewer’s Report

Selective liquid-phase oxidation of toluene with molecular oxygen catalyzed by Mn3O4 nanoparticles immobilized on CNTs under solvent-free conditions

Manuscript ID: catalysts-809129

In this manuscript, the authors have investigated the carbon nanotubes (CNTs) supported Mn3O4 catalysts for the selective liquid-phase oxidation of toluene with molecular oxygen. They reported that Mn3O4 nanoparticles loaded on CNTs performed better compared with pristine Mn3O4 or CNTs. The results presented in this manuscript could be publishable after the following revisions.

  • The authors should have to provide the XRD of catalysts at different carbon nanotubes loadings.
  • Calculate the crystallite size of the samples.
  • Page 4, “In addition to the Mn3O4 diffraction peak observed at 659 cm-1 [37], D-band diffraction peaks reflecting…………”

       However, the Raman peaks are not obtained from the diffraction.

  • The authors should have to fit the Mn 2p XPS spectra to confirm the surface Mn oxidation states.
  • It would be better if authors can provide the EDS mapping to know the distribution of Mn species on CNTs.
  • Improve the discussion on structure-activity relationships.
  • Enhance the conclusions section.

Reviewer 4 Report

Comments on the manuscript entitled “Selective liquid-phase oxidation of toluene with molecular oxygen catalyzed by Mn3O4 nanoparticles immobilized on CNTs under solvent-free conditions” by Feng and Zeng (catalysts-809129).

GENERAL COMMENT

The submitted manuscript deals with catalytic oxidation of toluene on Mn3O4 supported on carbon nanotubes. The paper presents a group of interesting catalyst characterization tests and activity tests. This paper could be publishable, however the following comments and suggestions should be addressed in a revised manuscript before it can be reconsidered for publication.

(1) It is known that, during the catalytic oxidation of several compounds, polymeric compounds can be formed and promoted by acidic sites on the catalyst, which can be adsorbed at the surface of the catalyst and decrease its activity. Could you explain better this aspect in yours results?.

(2) Could give the accuracy of your data in the figures 5 to 9?

(3) In terms of being able to observe the stability of the catalyst, it would be convenient to determine the BET area after the reuse of the catalyst.

(4) In the characterization section, could the authors show the graphs of adsorption isotherms, as well as the pore size distribution?

Other check:

(5) Please check the whole manuscript in order to correct English grammar mistakes.

(6) What is the novelty of this work?. The authors should emphasize this fact.

This paper need a major revision and suggestions should be addressed in a revised manuscript before it can be reconsidered for publication.

Reviewer 5 Report

Some problems need to be taken care of in this paper. the abstract and conclusions sections should quantitatively summarize the findings of this work. 58, take out it can be found that. 86, what does higher mean? More details are needed for the characterization section. Figure 1, identify all of the peaks. 146, take out t can be seen that. 166, take out it was found that. 174, take out it can be observed that. 182, take out it can be hypothesized that. Table 1, what do the - entries mean? Put this is the table caption. 187, take out it can be seen that. 192, take out it can be observed that. 202, take out it can be seen that. Table 2, check significant figures. 217, take out it can be seen that. Figure 5 use error bars. 249, take out it can be hypothesized that. Figure 7, use error bars. 257, what do small, small, small, low mean? Figure 9 use error bars. 

Round 2

Reviewer 4 Report

The authors have correctly made the reviewer's suggestions. The manuscript can be published.